# Physical Activity Patterns among Individuals with Prediabetes or Type 2 Diabetes across Two Years—A Longitudinal Latent Class Analysis

**DOI:** 10.3390/ijerph19063667

**Published:** 2022-03-19

**Authors:** Jenny Rossen, Maria Hagströmer, Kristina Larsson, Unn-Britt Johansson, Philip von Rosen

**Affiliations:** 1Department of Health Promoting Science, Sophiahemmet University, Lindstedtsvägen 8, 114 86 Stockholm, Sweden; maria.hagstromer@ki.se (M.H.); kristina.larsson@shh.se (K.L.); unn-britt.johansson@shh.se (U.-B.J.); 2Department of Neurobiology, Care Sciences and Society, Division of Physiotherapy, Karolinska Institutet, Alfred Nobels Allé 23, 141 52 Huddinge, Sweden; philip.von.rosen@ki.se; 3Academic Primary Care Center, Region Stockholm, Solnavägen 1E, 104 31 Stockholm, Sweden; 4Department of Clinical Science and Education, Södersjukhuset, Karolinska Institutet, Sjukhusbacken 10, 118 83 Stockholm, Sweden

**Keywords:** accelerometery, latent class, physical activity pattern, sedentary behaviour, time-use, trajectories

## Abstract

Background: This study aimed to identify distinct profiles of physical activity (PA) patterns among individuals with prediabetes or type 2 diabetes participating in a two-year PA trial and to investigate predictors of the profiles. Methods: Data (*n* = 168, collected 2013–2020) from the cohort of a randomized trial aimed at increasing PA in individuals with prediabetes and type 2 diabetes were used. PA and sedentary behaviours were assessed by waist-worn ActiGraph GT1M accelerometers at baseline and at 6, 12, 18 and 24 months. Fifteen PA and sedentary variables were entered into a latent class mixed model for multivariate longitudinal outcomes. Multinominal regression analysis modelled profile membership based on baseline activity level, age, gender, BMI, disease status and group randomisation. Results: Two profiles of PA patterns were identified: “Increased activity” (*n* = 37, 22%) included participants increasing time in PA and decreasing sedentary time. “No change in activity” (*n* = 131, 78%) included participants with no or minor changes. “Increased activity” were younger (*p* = 0.003) and more active at baseline (*p* = 0.011), compared to “No change in activity”. No other predictor was associated with profile membership. Conclusions: A majority of participants maintained PA and sedentary patterns over two years despite being part of a PA intervention. Individuals improving PA patterns were younger and more active at baseline.

## 1. Introduction

Physical activity (PA) has several beneficial effects on cardiometabolic health, and support for regular PA behaviour constitutes a cornerstone in type 2 diabetes prevention and treatment [1]. Regular physical activity may enhance mental health and glycaemic management and may reduce risk factors for cardiovascular disease in people with prediabetes and type 2 diabetes [2,3,4]. Regular exercise improves insulin sensitivity, lipids and blood pressure, and breaking up sitting may improve postprandial glucose and insulin levels [2,3,4]. However, behaviours such as PA are challenging to change and people tend to sustain their PA pattern over time [5,6,7]. Moreover, individuals participating in PA interventions most likely respond differently to the intervention [8]. 

Most PA intervention studies present either the total volume of PA (e.g., accelerometer counts or step counts) over a day or a week or investigates different intensities of PA separately while adjusting for time in other intensities [9]. Likewise, many interventions have a focus on sedentary behaviour (SB) [10], but few intervention studies apply a time-use perspective and consider the interrelated relationship between SB and different intensities of PA. It is problematic to study changes in PA and SB when the full pattern of PA is not regarded [11]. It may be the case that e.g., time in higher intensity PA is compensated for by spending more time in SB [12]. Therefore, it has been proposed to consider the relative time of different intensities across a day [13]. 

Insight into how the PA and SB patterns change during interventions is required to understand responsiveness to interventions and predictors for behaviour change [8]. Enhanced knowledge of PA and SB patterns and predictors for both beneficial and unfavourable change is required to improve and tailor the support for PA and reduced SB in diabetes care. 

A latent profile analysis has been applied to identify cross-sectional patterns of PA and SB among different adult populations and to link profile groups to health outcomes and mortality [14,15,16,17,18]. Previous studies have also applied latent profile analysis on longitudinal data to study trajectories of self-reported PA and predictors of PA patterns, e.g., on individuals with prediabetes [19], on general adult populations [6], among older adults [7] and in a worksite PA intervention [20]. To our knowledge, no study has applied latent profile analysis based on objectively measured PA to study changes in PA patterns during a PA intervention on adult populations with prediabetes or type 2 diabetes.

In previous studies, we reported the effects of the Sophia Step Study, an RCT using pedometers and counselling as tools to promote PA in diabetes care [21,22]. The results revealed an intervention effect on the maintenance of moderate to vigorous PA (MVPA) but a weak effect on daily steps [22]. This current study applies secondary analyses to further explore changes in PA behaviour across all participants. 

The aim of this study was to identify distinct profiles of PA pattern in individuals with prediabetes or type 2 diabetes participating in a two-year PA trial. A further aim was to investigate predictors of the PA pattern profiles.

## 2. Materials and Methods

This study uses longitudinal data from the cohort of a randomised controlled trial that aimed to evaluate the effects of self-monitoring daily steps as a motivational tool for being physically active on a regular basis among individuals with prediabetes or type 2 diabetes [23]. In brief, participants were randomised to (I) a multi-component intervention including a pedometer and a webpage to register and track daily steps, twelve group counselling sessions and nine individual counselling sessions for two years, (II) a single component intervention including pedometers and a webpage to register and track daily steps, and (III) a control group receiving standard care (meeting a general practitioner and a diabetes specialist nurse twice yearly and receiving advice to be regularly physically active). Inclusion criteria: HbA1c > 39 mmol/mol or fasting glucose > 5.6 mmol/L; 40–80 years of age and able to communicate in Swedish. Exclusion criteria: have suffered from a myocardial infarction in the past 6 months; serum creatinine > 140 mmol/L; diabetic foot ulcer or risk of ulcer (severe peripheral neuropathy); newly been prescribed insulin (<6 months); additional disease prohibiting physical activity; have suffered repeated hypoglycaemia or severe hypoglycaemia in the past 12 months; very physically active according to the Stanford Brief Activity Survey [24] and have no access to internet. 

Data were collected between April 2013 and January 2020. Approximately 400 individuals were assessed for eligibility and informed about the study at two primary care centres in central Stockholm and one in the south of Sweden. In total, 188 persons met the inclusion and exclusion criteria and consented to participation. Of these 188, 20 were excluded because of lack of PA data. Thus, the final sample comprised 168 eligible participants. All participants agreed upon participation by signing a written informed consent. The Regional Ethical Review Board has approved the study (Dnr.2012/1570-31/3 and 2015/2075-32).

### 2.1. Physical Activity and Sedentary Behaviour

PA and SB were assessed by a waist-worn ActiGraph GT1M accelerometer (ActiGraph, Pensacola, FL, USA) worn for seven consecutive days at baseline and at 6, 12, 18 and 24 months. Procedures for data collection and processing have been reported previously [25]. Participants providing data for ≥3 days and ≥10 h per day were included in the analyses [26]. Zero values of MVPA (*n* = 2) were imputedbased on maximum likelihood estimation [27]. Non-wear time was based on 90 min (min) of consecutive zero counts, with allowance of 2 min intervals for nonzero counts [28]. SB was defined as <100 counts/min [29], light intensity PA (LIPA) was defined as 100–1951 counts/min and MVPA was defined as ≥1952 counts/min [20]. An MVPA bout was defined as ≥10 min consecutive minutes of MVPA [30], and a sedentary bout was defined as ≥30 min consecutive minutes of SB. Number of days achieving the physical activity recommendations of at least 150 min/week of MVPA was calculated by averaging daily time in MVPA over the included days and multiplied by 7.

### 2.2. Demographic, Anthropometric, Disease and Medication Status

The following potential predictors were collected at baseline from medical records and through questionnaires: age, gender, BMI, hypertension, chronic obstructive pulmonary disease and other diseases (hyperlipidaemia, other cardiovascular disease, cancer during the past 5 years and inflammatory disease). BMI was calculated by the following formula: BMI = weight (kg)/height (m^2^).

### 2.3. Data Analysis

A total of 15 activity variables based on previous literature [13,31] were derived from the accelerometer data to capture a wide range of characteristics of PA and SB. These were time spent in different intensities (SB, LIPA and MVPA); variation in time spent for different intensities (standard deviation of time spent for SB, LIPA and MVPA); time of sedentary bouts; total time of MVPA bouts; number of sedentary and MVPA bouts; total counts (Counts); number of steps (Steps); and time spent on one behaviour relative to the two remaining behaviours (e.g., SB vs. LIPA, MVPA). The relative time spent on one behaviour was calculated as isometric log-ratio coordinates [32]. Since we use a 3-part composition (SB, LIPA and MVPA), three variables were derived representing the relative time in one behaviour (e.g., SB) relative to the average of the two other behaviours (e.g., LIPA and MVPA). The activity variables were calculated across the days in which the participants had at least 10 h of wear time and transformed into z-scores. Spearman correlations between the fifteen variables were used to identify multicollinearity, which resulted in the number of sedentary/MVPA bouts, total time of sedentary/MVPA bouts, total counts, number of steps, relative time for LIPA/MVPA and variation in time spent for SB/LIPA being excluded from the latent class mixed model. 

A latent class mixed model for multivariate longitudinal outcomes was conducted using R package “lcmm” [33]. Assumed correlated random effects for time were included, and models with 2 to 5 profiles were explored. The optimal number of profiles were evaluated using criteria described by Nylund et al. [34], and the final model was chosen based on the following fit statistics: (1) Akaike Information Criterion (AIC) and the Bayesian Information Criterion (BIC); (2) entropy values; (3) mean of posterior probability in each profile; and (4) meaningfulness of profile membership. The 2-profile model had the highest posterior probability and entropy value, compared to the other models, and was therefore chosen as the final model. The results of the different consecutive latent profile models with 2- to 5-profile solutions are presented in Table 1. Each participant was then assigned to one of the two profiles, based on the highest posterior probability of belonging in that profile. 

A logistic regression analysis was conducted to model profile memberships. The initial model was adjusted for gender, age and group randomisation. Possible predictors were BMI, hypertension (yes/no), chronic obstructive pulmonary disease (yes/no), other diseases (yes/no) and achieving the recommended guidelines of PA at baseline (≥150 min/week of MVPA). Based on the initial model, the independent variables were entered by backward elimination, and the final model was chosen based on information criteria (e.g., AIC and BIC). The final model was checked for linear relationships between the logit of the outcome and continuous variables and for intercorrelation between predictors and influential values. All analyses were conducted using the R statistical system version 3.5.2 (R Core Team 2021 Vienna, Austria).

## 3. Results

Two latent profiles were identified and labelled: Profile 1, “Increased activity” (*n* = 37, 22%), reflects participants that decreased time spent in SB and increased time spent in PA across time (e.g., increased numbers of daily steps, counts and time in MVPA); Profile 2, “No change in activity” (*n* = 131, 78%), reflects participants that showed no or small changes in SB and PA across time. In the “No change in activity” profile, no activity variable changed more than 0.5 *z*-value across time. Descriptions by profile membership are presented in Table 2.

In Figure 1, line plots by profile membership are presented across time for the activity variables for which the differences between the two profiles were distinct. The remaining line plots of the activity variables are presented in Appendix A: Appendix Figure Line Plots.

The logistic regression analysis showed that age and reaching 150 min of PA per week at baseline were significantly (*p* < 0.05) associated with profile membership (Table 3). Participants belonging to “Increased PA” were younger (*p* = 0.003) and achieved the recommended level of PA at baseline to a higher degree (*p* = 0.011) compared to “No change in PA”. Group randomisation, gender, BMI, hypertension, chronic obstructive pulmonary disease, and other diseases were not associated (*p* > 0.05) with profile membership. 

## 4. Discussion

To the best of our knowledge, this is the first study using latent profile analysis with a time-use approach of objectively measured PA to study PA pattern over time in a population with prediabetes and type 2 diabetes. Two profiles of activity patterns were identified. Most participants (78%) changed their PA pattern to a minor extent across the period and were classified into the profile “No change in activity”, and 22% changed positively and were classified into the profile “Increased activity”. Individuals in “Increased activity” were younger (*p* = 0.003) and achieved the recommended level of PA at baseline to a higher degree (*p* = 0.011), compared to “No change in activity”. The major difference in the pattern was for MVPA. While “No change in activity” kept their PA patterns steady, MVPA increased over time between the measurement time points for “Increased activity”.

Our results with mostly stable levels of PA over time are similar to recently published non-experimental studies applying latent profile analyses on self-reported PA among individuals with prediabetes [19], older adults [7], and among general populations [6]. Similar findings with mostly stable levels over time were also observed in a non-experimental study using objectively measured PA (daily steps) in an adult female population [35]. 

In this current trial, an important finding was that being physically active at baseline predicted belonging to “Increased activity”; thus, the already active individuals became more active. Comparable intervention studies applying profile groups in order to classify PA patterns are rare. A worksite PA intervention among adults applying self-reported PA reported contradictory findings with predominantly positive change from initially low PA levels [20]. A majority of participants (65%) in the intervention group were classified in the “increase from low PA” group, whereas 28% were classified as “stable moderate” and 17% as “decrease from high PA” [20]. Similar findings with a greater increase in self-reported PA among individuals with initially low levels of PA compared to individuals with initially high PA levels were reported in a study evaluating PA prescriptions among individuals at metabolic risk [36]. Conflicting findings between intervention studies may be explained by the intervention components as well as by characteristics of the study population and methods used to measure the studied behaviour. In this current study, a majority (58%) of the sample achieved recommended levels of PA at baseline, and we should perhaps not expect all of these individuals to increase PA levels further. Maintenance of the PA levels among these individuals is a relevant target. It is important to further explore resistance to change among the inactive individuals. It is known that inactive individuals report distinct barriers towards PA and require other support strategies compared with physically active individuals [37,38]. 

Younger age was another predictor of belonging to the group of “Increased activity” in this study. Mean age differed by 5 years between the profiles and it is unknown why younger individuals were more likely to increase time spent on PA and decrease sedentary time across two years. Previous research showed similar findings that older age predicted low [35] or stable [20] activity levels. The generally declining PA levels with age in older adults [6,39] along with a variety of determinants for PA among older adults [7] could be explanations. Various health-related factors that are linked to older age have been shown to predict an inactive or low-active trajectory of PA [6]. Supporting older adults in becoming more active possibly requires additional strategies rather than self-monitoring PA and counselling [40,41]. 

BMI and gender were not predictive of change in PA in this study. This is contradictory to earlier studies showing that individuals who are overweight and obese were less likely to change PA compared to individuals who are of normal weight [6,35,36]. Supporting PA may require different strategies for individuals who are obese, and, indeed, an approach tailored to the individual [42]. Male gender has been reported as a predictor for being active in general populations and for stable active and inactive PA patterns [3] and a more favourable PA pattern over time [43]. 

Intervention group allocation was not predictive of profile group membership in this study, which contradicts the findings from the study of the effects of the intervention [22]. Thus, the findings of this study highlight that factors other than the intervention were more important for the PA pattern. Initial PA levels and age predicted change, but BMI, gender and taking part in a PA intervention did not. However, numerous factors that may have influenced PA pattern, e.g., motivation, environmental factors and social support, were not evaluated in this study. Future PA intervention studies should explore individual trajectory changes and investigate possible influencing predictors and moderators in addition to intention-to-treat analyses. 

The major strength of this study is the use of repeated longitudinal data of objectively measured PA across two years. The use of latent class modelling to classify individuals according to change in PA and SB is an approach that allows for an investigation of patterns in the complex behaviour of PA. Fifteen separate variables were used to cover the pattern of the interrelated behaviours PA and SB. In addition, the use of a compositional data framework allowed for time-use application of the PA behaviours.

The weaknesses of this study are mainly due to the limited number of participants. Hence, the findings from the predictor analyses should be regarded as indications. The study did not regard predictors that may explain motivation to engage in physical activity or certain determinants of PA maintenance (e.g., self-efficacy), psychosocial factors, societal norms, or genetic or environmental factors that influence activity levels [44,45]. 

In summary, individuals with type 2 diabetes possess an elevated risk of atherosclerotic cardiovascular morbidity and mortality, and PA has several beneficial health effects that may potentially lower this elevated risk. Two profiles were identified, with a clear association between group membership and PA pattern over time. The profile “Increased activity” increased the absolute and relative time of MVPA across time, while the relative time of LIPA or SB decreased. In the “No change in activity” profile, no or small changes in terms of PA pattern were observed.

## 5. Conclusions

The majority of participants maintained their PA and SB patterns over two years regardless of whether they were part of a PA intervention or not. The group of individuals improving PA patterns were younger and more active at baseline.

## Figures and Tables

**Figure 1 ijerph-19-03667-f001:**
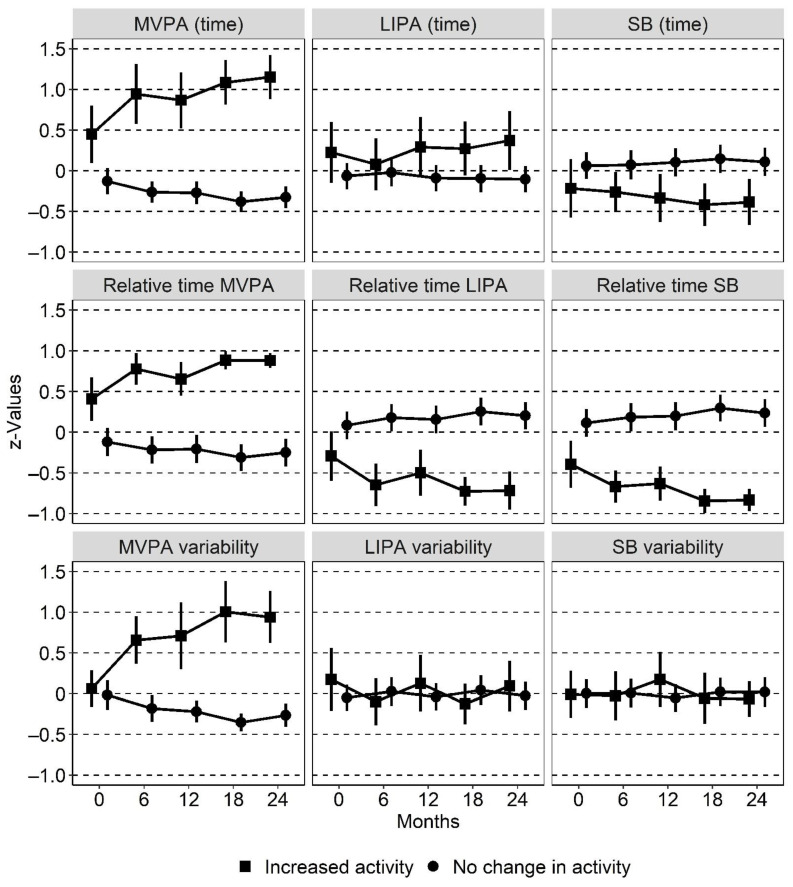
Line plots with 95% confidence intervals as error bars for PA and SB variables by profile membership across time. LIPA, light intensity physical activity; MVPA, moderate-to-vigorous physical activity; SB, sedentary behaviour. “Time” refers to absolute time for each behaviour, “relative time” refers to time in one behaviour relative to the two remaining behaviours and “variability” refers to the standard deviation of one behaviour across the days it was measured. All data are presented as *z*-values.

**Table 1 ijerph-19-03667-t001:** Fit indices of the 2- to 5-profile latent class mixed models.

Profiles	AIC	BIC	Lowest Mean Value of Posterior Probability in Each Profile	Entropy
2	9071	9137	0.83	0.70
3	9058	9143	0.51	0.63
4	9057	9141	0.43	0.53
5	9125	9219	0.25	0.12

AIC: Akaike information criterion; BIC: Bayesian information criterion.

**Table 2 ijerph-19-03667-t002:** Baseline characteristics of the study sample for all participants and by profile, with mean (SD) or number (%).

	All Participants (*n* = 168)	Increased Activity (*n* = 37)	No Change in Activity (*n* = 131)
Female (%)	66 (39)	15 (41)	51 (38)
Age (SD)	64.3 (7.7)	60.6 (8.4)	65.3 (7.2)
BMI (SD)	30.1 (4.7)	29.5 (4.2)	30.3 (4.8)
Randomisation (%)			
Multicomponent intervention	61 (36)	14 (38)	47 (36)
Single component intervention	51 (30)	15 (41)	36 (27)
Control group	56 (33)	8 (22)	48 (37)
Hypertension (%)	127 (76)	24 (65)	103 (79)
Chronic obstructive pulmonary disease (%)	16 (10)	1 (3)	15 (12)
Other disease * (%)	31 (19)	8 (22)	23 (18)
Achieving ≥150 min of moderate to vigorous per week at baseline (%)	92 (55)	29 (78)	63 (48)

* Hyperlipidaemia, other cardiovascular disease, cancer during the past 5 years or inflammatory disease.

**Table 3 ijerph-19-03667-t003:** Logistic regression analysis to model profile membership.

Model *	OR (95% CI)	Standard Error	*p*-Value
Gender (reference female)			
Male	1.08 (0.48–2.47)	0.41	0.853
Age	0.92 (0.87–0.97)	0.03	0.003
Randomisation group (reference Multicomponent intervention) Single component intervention Control group	1.70 (0.69–4.30) 0.54 (0.18–1.51)	0.47 0.54	0.254 0.253
Achieving 150 min/week moderate to vigorous PA at baseline	3.18 (1.34–8.24)	0.46	0.011

OR: odds ratio; CI: confidence interval. * Intercept b = 2.97.

## Data Availability

The data presented in this study are available from the corresponding author upon request. The data are not publicly available since data can be traced back to the study participants. This means that, according to Swedish and EU data legislation, access can only be made available upon reasonable request.

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
