# Peer review of "Physical Activity Patterns among Individuals with Prediabetes or Type 2 Diabetes across Two Years—A Longitudinal Latent Class Analysis"

_ijerph, 2022, doi:10.3390/ijerph19063667_

Round 1
Reviewer 1 Report
The topic of this article is highly interesting and relevant as it is important to understand what are the determinants (or predictors) of physical activity (PA) over a long period of time (in this study, 2 years). However, a few things could be improved.
Introduction
1) The introduction is a bit short. More studies could be cited to justify why PA is important for adults with prediabetes and type 2 diabetes (e.g., benefits of regular PA for this specific population).
2) In the introduction, the authors mention that the Sophia Step Study—the study on which the present results are based on—had a significant impact on maintenance of moderate to vigorous PA (MVPA), but a weak effect on daily steps. This seems to contradict the results of the present study whereby belonging to the multicomponent or single component intervention vs. the control group had no significant effect on PA.
Materials and Methods
3) The authors mention that “very physically active according to the Stanford Brief Activity Survey” participants were excluded from the study. However, later in the paper (e.g., Table 2), it is mentioned that 55% of all participants already achieved at least 150 min. of MVPA/week at baseline. Again, this seems contradictory. I am not sure how adults already achieving at least 150 min. of MVPA/week could benefit from an intervention aimed at increasing PA.
Results
4) I have a problem with the labels used for the two latent profiles. The term “responders” is usually used to refer to people who completed the intervention (similar to “completers”). Maybe the term “individuals who increased PA” would be more suitable? My main problem is with the term “maintainers”. In the literature on PA maintenance, “maintenance” usually refers to previously sedentary individuals who increased their PA level after a PA intervention or on their own who are still regularly exercising (e.g., at least 150 minutes of MVPA/week) for a given period of time (e.g., at least 6 months) (see Amireault, Godin & Vézina-Im, 2013, Determinants of PA maintenance: A systematic review and Meta-Analyses, Health Psychol Rev, 7, 55-91; Marcus et al., 2000, PA Behavior Change: Issues in Adoption and Maintenance, Health Psychol, 19, 32-41). Therefore using the term “maintainers” is problematic for people who maintained a sedentary lifestyle. Maybe the term “individuals with no change in PA” would be more suitable? Also, a difference should be made between already physically active people who did not change their PA level over 2 years, and sedentary people who did not change their PA level over 2 years. The first category of people are likely to benefit from maintaining their physically active lifestyle while the second category of people are at risk for health issues (e.g., complications from their prediabetes and/or type 2 diabetes) for staying sedentary.
Figure 1
5) Please add a footnote at the bottom of this figure to explain what is meant by each category: “(time)”, “relative time”, and “variability”.
Discussion
6) I would add that a weakness of this study is that no psychosocial variables or theoretical constructs related to PA maintenance, such as self-efficacy or intention, were measured (see Amireault, Godin & Vézina-Im, 2013, Determinants of PA maintenance: A systematic review and Meta-Analyses, Health Psychol Rev, 7, 55-91; Marcus et al., 2000, PA Behavior Change: Issues in Adoption and Maintenance, Health Psychol, 19, 32-41; Rhodes & Sui, 2021, PA Maintenance: A Critical Narrative Review and Directions for Future Research, Front Psychol).
Non-published Material
7) I am not sure the unpublished material is relevant, since it is written in another language than English.
Author Response
Response to reviewer 1 comments
Point: The topic of this article is highly interesting and relevant as it is important to understand what are the determinants (or predictors) of physical activity (PA) over a long period of time (in this study, 2 years). However, a few things could be improved.
Response: Thank you for your valuable comments
Introduction
Point 1: The introduction is a bit short. More studies could be cited to justify why PA is important for adults with prediabetes and type 2 diabetes (e.g., benefits of regular PA for this specific population).
Response: We have expanded the background and added benefits of PA for adults with prediabetes and type 2 diabetes. Page 1, line 39-43.
Point 2: In the introduction, the authors mention that the Sophia Step Study—the study on which the present results are based on—had a significant impact on maintenance of moderate to vigorous PA (MVPA), but a weak effect on daily steps. This seems to contradict the results of the present study whereby belonging to the multicomponent or single component intervention vs. the control group had no significant effect on PA.
Response: The results may seem contractionary and are perhaps somewhat disappointing. There were indeed statistically significant differences between the intervention groups and the control group when comparing the groups. These differences were rather small though, and this current study shows belonging to one of the intervention groups did not predict profile group membership. We have clarified the discussion somewhat (page 9, Line 256-258). We discuss that the drivers of the PA pattern may be other predictors and mediators rather than the intervention, as for example age and initial levels of PA that were shown to be predictive in this study.
Materials and Methods
Point 3: The authors mention that “very physically active according to the Stanford Brief Activity Survey” participants were excluded from the study. However, later in the paper (e.g., Table 2), it is mentioned that 55% of all participants already achieved at least 150 min. of MVPA/week at baseline. Again, this seems contradictory. I am not sure how adults already achieving at least 150 min. of MVPA/week could benefit from an intervention aimed at increasing PA.
Response: Very physically active according to the Stanford Brief Activity Survey is comparable to a PA dose that is above the recommended levels (at that time the recommendations were 150 min/week in MVPA). This was applied to exclude highly active individuals. Thus, with the intervention we did reach people with sufficient PA levels as well as inactive people. Active people in this age group may need support from the diabetes care provider to keep their PA levels over time and is a relevant target group in primary care. However, this makes the interpretation of the findings more challenging. An interesting, and important, finding of this study was that being active at baseline predicted belonging to the group of responders (now changed to “Increased activity”). We have added to the discussion that this was an important finding (page 7, Line 219) to highlight this result.
Results
Point 4: I have a problem with the labels used for the two latent profiles. The term “responders” is usually used to refer to people who completed the intervention (similar to “completers”). Maybe the term “individuals who increased PA” would be more suitable? My main problem is with the term “maintainers”. In the literature on PA maintenance, “maintenance” usually refers to previously sedentary individuals who increased their PA level after a PA intervention or on their own who are still regularly exercising (e.g., at least 150 minutes of MVPA/week) for a given period of time (e.g., at least 6 months) (see Amireault, Godin & Vézina-Im, 2013, Determinants of PA maintenance: A systematic review and Meta-Analyses, Health Psychol Rev, 7, 55-91; Marcus et al., 2000, PA Behavior Change: Issues in Adoption and Maintenance, Health Psychol, 19, 32-41). Therefore using the term “maintainers” is problematic for people who maintained a sedentary lifestyle. Maybe the term “individuals with no change in PA” would be more suitable? Also, a difference should be made between already physically active people who did not change their PA level over 2 years, and sedentary people who did not change their PA level over 2 years. The first category of people are likely to benefit from maintaining their physically active lifestyle while the second category of people are at risk for health issues (e.g., complications from their prediabetes and/or type 2 diabetes) for staying sedentary.
Response: Thank you for the clarifications. We agree that the naming of the profile groups may be misinterpreted and have changed them to “increased activity” and “no change in activity”, which we think assists in the understanding and interpretation.
Figure 1
Point: 5: Please add a footnote at the bottom of this figure to explain what is meant by each category: “(time)”, “relative time”, and “variability”.
Response: We have added explanation for “time”, “relative time”, and “variability”.
Discussion
Point 6: I would add that a weakness of this study is that no psychosocial variables or theoretical constructs related to PA maintenance, such as self-efficacy or intention, were measured (see Amireault, Godin & Vézina-Im, 2013, Determinants of PA maintenance: A systematic review and Meta-Analyses, Health Psychol Rev, 7, 55-91; Marcus et al., 2000, PA Behavior Change: Issues in Adoption and Maintenance, Health Psychol, 19, 32-41; Rhodes & Sui, 2021, PA Maintenance: A Critical Narrative Review and Directions for Future Research, Front Psychol).
Response: We have expanded this section and added a few weaknesses, page 9 Line 274-275, as suggested by the Reviewer.
Non-published Material
Point 7: I am not sure the unpublished material is relevant, since it is written in another language than English.
Response: The unpublished material was the decision for ethical approaval which was requested from the journal.
Reviewer 2 Report
Based on longitudinal data collected previously, they aimed to identifiy and analyse profiles of physical acitivty in pre-and diabetic patients during 2 years.
Line 169: I guess that it may be an additional setence that it is not necessary for the manuscript.
Line 186: I guess that the information between parenthesis is not necessary
I would like to understand the clinical relevance of this study and the diabetes context is missing. The importance of physical activty in this population should be stated.
Can you establish a dose-response between levels of physical activity and clinical profile in this specific population?
Author Response
Response to reviewer 2
Point 1: Line 169: I guess that it may be an additional setence that it is not necessary for the manuscript.
Response: We agree and have removed that sentence.
Point 2: Line 186: I guess that the information between parenthesis is not necessary
Response: We have removed the information in parenthesis.
Point 3: I would like to understand the clinical relevance of this study and the diabetes context is missing. The importance of physical activty in this population should be stated.
Can you establish a dose-response between levels of physical activity and clinical profile in this specific population?
Response: We have clarified the diabetes context and added the importance of physical activity in this population (see introduction and the last section of Discussion). It is not common to describe a dose-response in this case, as the “exposure” are based on two profiles and not a continuous variable. However, we describe how the profiles changes over time, which could be seen as a certain measure of “dose-response”.
Round 2
Reviewer 1 Report
The authors did an excellent job at integrating my comments and the manuscript has improved as a result.